# Green Salad Intake Is Associated with Improved Oral Cancer Survival and Lower Soluble CD44 Levels

**DOI:** 10.3390/nu13020372

**Published:** 2021-01-26

**Authors:** Elizabeth Bradford Bell, Isildinha M. Reis, Erin R. Cohen, Turki Almuhaimid, Drew H. Smith, Faisal Alotaibi, Claudia Gordon, Carmen Gomez-Fernandez, W. Jarrard Goodwin, Elizabeth J. Franzmann

**Affiliations:** 1Department of Otolaryngology, University of Miami Miller School of Medicine, Miami, FL 33136, USA; e.bradfordbell@unmc.edu (E.B.B.); erin.cohen@jhsmiami.org (E.R.C.); dralmohimmed@yahoo.com (T.A.); dxs1439@med.miami.edu (D.H.S.); cgordon2@med.miami.edu (C.G.); wgoodwin@med.miami.edu (W.J.G.); 2Sylvester Comprehensive Cancer Center, University of Miami Miller School of Medicine, Miami, FL 33136, USA; ireis@med.miami.edu; 3Department of Oral and Maxillofacial Surgery and Diagnostic Sciences, College of Dentistry, Prince Sattam bin Abdulaziz University, Alkharj 16245, Saudi Arabia; fisalfm@gmail.com; 4King Fahad Specialist Hospital, Dammam 32253, Saudi Arabia; 5Department of Pathology, University of Miami Miller School of Medicine, Miami, FL 33136, USA; cgomez3@jhsmiami.org

**Keywords:** diet, nutrition, salivary biomarkers, cancer outcomes, head and neck cancer

## Abstract

Deficiencies in fruit and vegetable intake have been associated with oral cancer (oral cavity and oropharyngeal). Salivary rinses contain measurable biomarkers including soluble CD44 (solCD44) and total protein, which are known markers of oral cancer risk. This study investigates the effect of nutritional factors on solCD44 and protein levels to evaluate oral cancer risk and survival. We evaluated solCD44 and protein levels from 150 patients with oral and oropharyngeal squamous cell carcinoma and 150 frequency-matched controls. We subsequently characterized the effect of food group consumption and these biomarkers on progression-free survival (PFS) and overall survival (OS). Patients reported eating fewer servings of salad (*p* = 0.015), while controls reported eating fewer servings of potatoes (*p* < 0.001). Oral cancer patients who consumed at least one serving per week of green salad were found to have significantly lower CD44 levels than those who ate salad less frequently (mean of log_2_[solCD44]1.73 versus 2.25, *p* = 0.014). Patients who consumed at least one serving per week of “salad or other vegetables” had significantly longer PFS (median 43.5 versus 9.1 months, *p* = 0.003, adjusted hazard ratio (HR) = 0.39 *p* = 0.014) and OS (median 83.6 versus 10 months, *p* = 0.008, adjusted HR = 0.04 *p* = 0.029). These findings suggest that dietary factors, namely greater green salad and vegetable intake, may be associated with lower CD44 levels and better prognosis in oral cancer patients.

## 1. Introduction

Each year, approximately 650,000 people are diagnosed with cancers arising in the head and neck worldwide [1]. In the United States, head and neck cancer accounts for 3% of malignancies, affecting 53,000 patients and leading to the death of 10,800 patients each year [2]. Squamous cell carcinoma represents the most common type of head and neck cancer, and carcinogenesis is primarily attributed to environmental factors such as smoking, alcohol consumption, human papillomavirus (HPV) infection, Epstein–Barr virus (EBV) infection, chronic periodontitis, occupation, genetics, and poor diet [3]. While smoking and tobacco have historically been associated with head and neck cancer, incidence of HPV-related head and neck cancer is rising. Oral cancer associated with HPV is distinct from smoking-related cancer in that it tends to affect a younger, healthier population, has a better prognosis, and is more commonly found in the oropharynx as opposed to the oral cavity [4].

Prolonged exposure to these risk factors in a genetically susceptible patient can lead to the transformation of normal cells and, eventually, to the development of cancer through the formation of carcinogens. Certain nutrients, namely vegetables, fruits, and tea, have been shown to decrease carcinogenic activity through the production of antioxidants [5]. Dietary deficiencies primarily related to a lack of adequate fruit and vegetable intake are thought to contribute to up to 60% of oral cavity, pharyngeal and esophageal malignancies in developing countries [6]. A similar relationship between dietary intake and head and neck cancer risk has been identified in industrialized countries as well. In a large prospective study conducted using the National Institutes of Health-AARP Diet and Health cohort, an inverse relationship between fruit and vegetable intake and head and neck cancer risk was established. In addition, the authors found a stronger association with vegetable intake compared to fruit intake [7]. Moreover, an additional prospective cohort study from the United States found that low fruit intake was associated with decreased survival in univariable analysis [8]. The European Investigation into Cancer and Nutrition conducted a prospective study that included patients from seven European countries; this group demonstrated a significant inverse relationship between fruit and vegetable intake and head and neck cancer risk [9]. A separate prospective study from Spain showed that vegetable intake was associated with improved prognosis in multivariable analysis [10]. The 2018 World Cancer Research Fund (WCRF) Diet and Cancer Report summarized that there is evidence to support a causal relationship between increased intake of non-starchy vegetables and decreased risk of head and neck cancer. However, the evidence is limited, necessitating further research in this area [11].

Cancer stem cells (CSCs) are thought to play an important role in tumorigenesis through their capacity for initiation, progression, recurrence, and resistance to chemo and radiation therapy [12]. CD44 is a marker for CSCs and has proven to be a reliable indicator for detecting head and neck squamous cell carcinoma (HNSCC), and it can be measured simply and inexpensively [13,14]. As a transmembrane surface glycoprotein, it serves as an adhesion molecule and interacts with cellular components such as hyaluronan, tyrosine kinases, and other cytoskeletal elements. Deviant expression of various CD44 isoforms can result in extension and metastases of head and neck cancer [15,16,17,18]. CD44 also interacts with and is expressed by immunologic cells, particularly T cells [19]. CD44 is a marker for CSCs and may be a reliable indicator of head and neck squamous cell carcinoma (HNSCC) that can be measured simply and inexpensively [13,14]. In a prior publication of our case-control study, we demonstrated that a higher level of the combination of soluble CD44 (solCD44) and salivary total protein in oral rinses was significantly associated with worse prognosis in HNSCC [14,20,21]. The purpose of this analysis was twofold. First, we use the original case-control design of our study to evaluate whether consumption of fewer servings of fruits and vegetables was associated with higher solCD44 and salivary total protein levels in oral cancer patients as well as controls. Second, we investigate the effects of diet (fruit and/or vegetable intake) on progression-free survival and overall survival of oral cancer cases, controlling for the two markers and other potential predictors. 

## 2. Materials and Methods

### 2.1. Patient Enrollment

In total, 150 newly diagnosed oral cancer patients from Jackson Memorial Hospital (JMH) and University of Miami Hospital and Clinics (UM) and 150 healthy controls were enrolled according to the case-control study protocol approved by the University of Miami Institutional Review Board between 2007 and 2012 (Table 1). Controls were frequency matched to cases for age, gender, race, ethnicity, tobacco use, alcohol use, and socioeconomic status (SES). Oral cancer cases included newly diagnosed, previously untreated subjects with HNSCC involving the oropharynx (OP) and oral cavity (OC). Control subjects were identified from family medicine and internal medicine clinics and chosen, prior to testing, so that the key covariates (age, race, tobacco and alcohol use, etc.) in the control group were not significantly different from the covariates in the case group. All subjects were recruited equally from UM, a private university hospital system serving mostly insured, Caucasian patients and JMH, a county hospital system serving primarily low-income patients and a large minority population. All subjects completed a questionnaire, including demographics and oral cancer risk factors. SES was determined by income, education, and employment. For cases, data on tumor characteristics and outcomes were extracted from medical records. The 7th edition of the AJCC cancer staging guidelines were used, given that the 7th edition was the most recent at the time of study enrollment and initial data collection [22]. Controls with lesions suspicious for oral cancer were excluded, as were HIV-positive or pregnant individuals. Exclusion decisions were blinded to marker level results.

### 2.2. Questionnaire

Participants in this case-control study completed a lifestyle questionnaire, which included questions about fruit and vegetable consumption. These questions noted frequency of intake with respect to drinking juices (fruit juices such as orange, apple, lemonade, grapefruit, tomato, or others), or eating fruit (not containing juice), green salads, potatoes (not including French fries, fried potatoes, or potato chips), carrots, and servings of other vegetables. The multiple-choice categories for servings consumed were as follows: 1–2, 3–4, or >4 per day; <1, 1–3, or 4–6 per week; 1–3 per month; 1–2, 3–4, or 5–6, or 5–10 per year; and never. We further separated frequency of intake into three categories (4–6 per week or more, 1–3 per week, or <1 per week or never) and into two categories (1–3 per week or more versus <1 per week or never). These data were captured during the enrollment period, which for cancer cases was close to the initial diagnosis. 

### 2.3. Oral Rinse Assays

Oral rinses were collected using a previously published method that samples the oral cavity and oropharynx [13,14,20,23]. Levels of solCD44 (normal and variant isoforms) were measured using a sandwich ELISA assay (eBioscience, San Diego, CA, USA), with previously published modifications [13,14,20,23]. We performed the DC protein assay (Bio­Rad Laboratories, Hercules, CA, USA) according to the manufacturer’s protocol using saliva samples prepared as previously published [13,14,20,23]. Each sample was tested in duplicate, and average values were reported. The technician was blinded to disease status.

### 2.4. HPV and CD44 Status

Formalin-fixed and paraffin-embedded specimens were retrieved from cases, where available (*n* = 79). HPV status was assessed using p16^INK4A^ immunohistochemistry (IHC), which is an accepted surrogate marker for HPV [24,25,26]. p16^INK4A^ was performed according to the manufacturer’s IHC protocol on 68 specimens (BD Bioscience, San Jose, CA, USA). Additionally, HPV status was already available in 11 cases (IHC, *n* = 10 or in situ hybridization, *n* = 1). All specimens were reviewed by a pathologist (CG), who was blinded to the patient’s clinical data. p16^INK4A^ expression was scored as positive if strong and diffuse nuclear and cytoplasm staining was present in ≥70% of the tumor specimen [26]. IHC was also used to confirm the presence of CD44 in tissue specimens, as documented in a previous publication [27].

### 2.5. Statistical Analysis

Patient groups were compared with respect to the distribution of potentially important categorical covariates using the chi-square test or Fisher’s exact test. Continuous variables were analyzed using Student t-test or analysis of variance (ANOVA) followed by Fisher’s least-significant-difference test for pairwise mean comparison. Data on solCD44 were log2-base transformed (Log_2_[solCD44]) to better approximation to normal distribution, as was done in previous publications [27,28]. Progression-free survival (PFS), which was defined as the elapsed time from date of diagnosis to progression or death, was evaluated in 137 cases. Overall survival (OS), which was defined as the elapsed time from date of diagnosis to death from any cause, was evaluated in all 150 oral cancer cases. PFS and OS curves were estimated by the Kaplan–Meier method and compared using the logrank test. Univariable and multivariable Cox proportional hazards regression analyses were performed to evaluate effect of nutritional variables on PFS and OS. Hazard ratio (HR) estimates with corresponding 95% confidence intervals (CIs) and *p*-values (*p*) derived from Cox regression models are reported. Multivariable models included adjustment for the following factors: HPV status, T-stage, solCD44, protein, age, race, ethnicity, gender, smoking history, drinking habits, SES, oral heath score, teeth removed, and gargle. Since HPV status (by P16 staining) was not available (NA) for a large number of HNSCC cases (71, 47.3%), “NA” was treated as a separated category in analyses or models including HPV (P16) status. Statistical analyses were performed using SAS version 9.4 (SAS Institute, Inc., Cary, NC, USA) with *p* ≤ 0.05 considered statistically significant.

## 3. Results

### 3.1. Patient Characteristics 

Patient characteristics are shown in Table 1. For both cases and controls, the majority of patients were less than 60 years old (cases: 56.0%; controls: 56.7%), male (80.7%; 78.7%), white race (82.6%; 79.7%), Hispanic ethnicity (51.3%; 62.0%), had poor/fair oral health (64.0%; 58.0%), and had a low socioeconomic status (66.7%; 60.0%). Our patient cohort had a high number of ever smokers, which was defined by a lifetime history of smoking at least 100 cigarettes (cases: 78.0%; controls: 78.7%), and less than half were moderate/heavy drinkers (48.0%; 43.3%). Of the 79 cases that underwent p16 staining, 39.2% were positive. Further definitions of included variables are in the Table 1 footnote and are consistent with stratification from a previous publication [28].

### 3.2. Association of Soluble CD44 and Protein with Nutrition

We compared cases and controls with respect to intake of various food groups as reported in the questionnaires. We evaluated intake of juices, fruit, green salads, potatoes, carrots, and servings of other vegetables. A significantly higher proportion of oral cancer cases ate fewer servings of salads than controls (23.9% vs. 12.7% reporting never or <1 servings/week, *p* = 0.015) (Appendix A). A significantly higher proportion of controls ate fewer servings of potatoes than oral cancer cases (35.0% vs. 15.3% in cases reporting never or <1 servings/week, *p* < 0.001). There were no other significant differences between cases and controls with respect to the food groups studied. 

CD44, a tumor-initiating marker, and total protein have been associated with HNSCC in case-control studies [12,13,14,23,29]. Since poor nutritional intake has been associated with increased risk of cancer, we evaluated whether differences in nutritional intake were associated with significant differences in solCD44 or protein levels. Consistent with prior work, mean log_2_[solCD44] was significantly elevated in the adjusted model of cases compared to controls (1.90 vs. 1.30, *p* < 0.0001, which corresponds to solCD44 values of 3.73 versus 2.46 ng/mL, respectively), as was mean total protein (0.94 vs. 0.81 mg/mL, *p* = 0.0243) (Appendix A). In univariable analysis, oral cancer cases who ate green salad one or more times per week were found to have significantly lower log_2_[solCD44] levels than those who ate salad less frequently (1.73 vs. 2.25, *p* = 0.014, which corresponds to solCD44 values of 3.32 versus 4.76 ng/mL, respectively) (Appendix A), suggesting that CD44 levels may vary with differences in risk such as salad intake. After adjusting for covariates, the difference reduced slightly, and it was not statistically significant (adjusted mean 1.74 versus 2.14, *p* = 0.081). No other statistically significant differences were seen with respect to marker levels and nutritional intake.

### 3.3. Progression-Free Survival (PFS) and Overall Survival (OS)

To determine whether nutritional factors were associated with prognosis, we evaluated PFS and OS in oral cancer with respect to intake of various food groups (Table 2 and Table 3). The estimates of effects of potential prognostic variables for PFS in univariable and multivariable analyses are shown in Table 2. Multivariable analysis included adjustment for the following factors: HPV status, T-stage, solCD44, protein, age, race, ethnicity, gender, smoking history, drinking habits, SES, oral heath score, teeth removed, and gargle. Cases who ate 1–3 servings of salad per week or more showed significantly better PFS on univariable analysis than cases who ate <1 serving of salad per week or never ate salad (HR = 0.51 [95%CI 0.31, 0.83]; *p* = 0.007). On multivariable analysis using adjustments above as depicted in model 1, the effect of salad intake was no longer statistically significant (HR = 0.72 [95%CI 0.42, 1.23]; *p* = 0.229). In subgroup analysis of PFS by p16 staining, no effect was observed in 45 p16 negative patients (HR = 0.98 [95%CI 0.43, 2.24]; *p* = 0.971). In 27 p16 positive patients, the effect of eating more green salads was not statistically significant (HR = 0.27 [95%CI 0.06, 1.16]; *p* = 0.079). In 65 patients who fell in the p16 NA category for unknown p16 status, patients who ate more green salads had significantly improved PFS (HR = 0.41 [95%CI 0.21, 0.82]; *p* = 0.011). Although “other vegetables” alone did not appear to impact PFS, more servings of “salads and other vegetables” among HNC patients resulted in significantly improved PFS on univariable analysis (HR = 0.39 [95%CI 0.20, 0.74]; *p* = 0.004). This relationship held true on multivariable model 2, which evaluated the effect of salad/other vegetables intake with the aforementioned adjustments (HR = 0.39 [95%CI 0.19, 0.83]; *p* = 0.014).

Similar results were observed with respect to OS (Table 3). In univariable analysis, eating 1–3 servings/week or more of salad as opposed to <1/week or never was a significant predictor of better OS (HR = 0.45 [95%CI 0.26, 0.77]; *p* = 0.003). The effect of eating more servings of salad was not statistically significant on multivariable model 1 (HR = 0.64 [95%CI 0.35, 1.16]; *p* = 0.141). In subgroup univariable analysis of OS by HPV-P16 category, the effect of eating more green salad on OS was not statistically significant in 45 p16-patients (HR = 0.67 [95%CI 0.27, 1.66]; *p* = 0.381) nor in 27 p16+ patients (HR = 0.25 [95%CI 0.05, 1.12]; *p* = 0.069). However, in 65 p16 NA patients, there was a statistically significant longer OS among those eating more green salad (HR = 0.43 [95%CI 0.20, 0.89]; *p* = 0.024). Our findings also demonstrated a significantly improved OS in patients who ate more green “salads and other vegetables” both on univariable (HR = 0.40 [95%CI 0.20, 0.81]; *p* = 0.011) and multivariable analysis model 2 (HR = 0.40 [95%CI 0.17, 0.91]; *p* = 0.029).

Kaplan–Meier curves for PFS and OS are depicted in Figure 1. Patients who consumed green salad more frequently had significantly improved PFS and OS, as shown in Figure 1A,B comparing three groups (PFS: *p* = 0.0160; OS: *p* = 0.0056), and in Figure 1C,D comparing two groups (PFS: *p* = 0.0061; OS: *p* = 0.0026). When combining salads and other vegetables (Figure 1E,F), oral cancer patients who consumed these at least 1–3 times per week had significantly improved PFS (median 43.5 months [95%CI 22.2, 94.9] versus 9.1 months [95%CI 5.0, 10.1], *p* = 0.0030) and OS (median 83.6 months [95%CI 54.1, not estimable] versus 10 months [95%CI 5.0, not estimable] *p* = 0.0084). There was no significant effect of other vegetables alone on PFS (*p* = 0.1855) and OS (*p* = 0.1902) (Figure 1G,H).

Plots in Figure 2 show estimated log hazard ratios (log HRs) for PFS and OS outcomes according to log2[solCD44] or protein as continuous variables. These estimates were obtained from Cox regressions with restricted cubic splines functions of log2[solCD44] or protein to allow no linear relationship between the continuous variable and the log HR of progression or death [30]. These plots confirm the cutoff points 8.1 for solCD44 and 1.05 for protein determined using SAS macro % FINDCUT [31] on the basis of minimum Wald test *p* value, and searching CD44 range 1.5 to 11 by 0.1 and protein range 0.5 to 1.7 by 0.05. Thresholds around 3 for log2 solCD44 (yielding value 8 for solCD44 = 8) and around 1 for protein define low and risk groups for progression and death.

## 4. Discussion 

Oral cancer is a debilitating and often deadly disease that has been historically associated with alcohol and tobacco consumption and more recently with HPV infection [32,33]. Quality of diet and its impact on risk of cancer development has been a burgeoning area of research. A number of large prospective studies have demonstrated a correlation between increased vegetable and fruit intake with decreased risk of cancer [7,8,9,10]. SolCD44 and salivary total protein have significant potential to be useful in both HNSCC screening but also as a means to predict the potential impact of behavioral change through dietary modification on cancer risk and cancer outcomes [12,34]. This case-control study demonstrates that subjects who consumed less green salad had higher CD44 levels and were more likely to be diagnosed with oral cancer. To our knowledge, this study is the first to demonstrate that CD44 levels are associated with both nutritional factors and cancer status.

Researchers continue to learn how various nutrients affect cancer growth and development. Similar to other studies, this study showed that green salad intake was inversely associated with oral cancer, while potato intake demonstrated a direct association. The International HN Cancer Epidemiology (INHANCE) Consortium pooled data included 22 case-control studies [35]. Food questionnaires were designed by each individual study. Overall vegetable intake (excluding potatoes) showed an inverse association with HNSCC (OR = 0.66 [95%CI 0.49–0.90], *p* trend = 0.01). Similar associations were observed for non-starchy vegetables, especially green vegetables and allium vegetables, but not for cruciferous vegetables. Consumption of green salad, lettuce, and fresh tomatoes more than seven times per week were associated with lower HNSCC risks compared to consumption less than once per week, similar to our findings. Potato intake was associated with an increased risk of HNSCC (OR = 1.24 [95%CI 1.05–1.46]), especially for fried potatoes (OR = 2.97 [95%CI 1.40–6.32]), which was also consistent with our study.

A large prospective cohort study done in Netherlands over a 20 year period included 120,852 participants who completed a 150-item food frequency questionnaire at baseline and then were followed for the development of HNSCC; findings demonstrated a significant association between total vegetable and fruit consumption and lower risk of HNSCC, which was particularly notable in oral cavity cancer [36]. Similarly, in a meta-analysis of 16 studies, consumption of vegetables and fruit was associated with a decreased risk of HNSCC overall. This study incorporated cancers of both the oral cavity and oropharynx, observing significant correlations with both fruit (combined odds ratio per portion: 0.51, [95%CI 0.40–0.65]) and vegetable intake (combined OR per portion: 0.50, [95%CI 0.38–0.65]), but it did not examine laryngeal cancer [37]. A significant association between beans and peas, peppers and tomatoes, and carrots showed decreased risk of head and neck cancer in a prospective observational study [7]. 

Our study results showed a trend (*p* = 0.079) toward lower fruit intake in cancer cases compared to controls; however, CD44 levels trended toward higher levels in the cancer cases who had higher fruit intake. This could be due to the types of fruits consumed. More detailed study on specific types of fruit and vegetable intake is needed to better understand our results in relationship to these other studies.

In the previous studies, the researchers concentrate on the prevention of HNSCC cancer by nutritional intake. There is a smaller body of work involving nutritional effects on the prognosis or overall survival in patients with HNSCC. An oral cancer patient’s nutritional intake in the months immediately prior to diagnosis may be affected by dysphagia or odynophagia. This may be a potential limitation, as nutritional intake recorded near the time of diagnosis may not reflect long-term or prior nutritional intake. In addition, recall bias should be acknowledged as an additional limitation, given that questionnaires were used for data collection. Patients pre- and post-treatment for HNSCC cancer suffer from malnutrition and anorexia, which affect prognosis and overall survival rate. Furthermore, baseline weight, body mass index (BMI), or exercise habits may have contributed to diet preferences, but these data were not captured during initial enrollment in 2008, as this project was not originally focused on nutritional outcomes. Regardless, results underscore the importance of a diet high in green salad and vegetables in the pretreatment period, although the foods may need to be pureed.

Here, we reviewed the prognostic impact of each of the nutrition variables. We found that patients who ate 1–3 servings of salad or more per week and those that ate more “salads and other vegetables” had significantly better PFS and OS. No significant relationship was identified between survival and each of the other variables, namely fruits, potatoes, carrots, juices, and “other vegetables” alone. When performing subset analyses by p16 status, only the p16 NA cohort showed significant improved PFS and OS (PFS: HR = 0.41, *p* = 0.011; OS: HR0.43, *p* = 0.024) for patients who consumed more salads. The p16-positive patients who consumed more salads appear to have improved PFS and OS compared to those who consumed less salad, but the difference between these groups was not statistically significant at the 5% significance level (PFS: *p* = 0.079; OS: *p* = 0.069). No significant difference was also observed for the p16 negative cohort. The significance of this finding is not fully understood, and further studies to evaluate this relationship are warranted.

## 5. Conclusions

In conclusion, our study shows soluble CD44, a cancer stem cell marker that can be measured simply and inexpensively, is elevated in oral cancer patients who eat less green salad. This work also confirms that green salad intake may be inversely associated with cancer cases and that both lower salad and “salad and other vegetables” intake may portend poorer prognosis. To our knowledge, this is the first study to show that increased green salad intake is associated with improved PFS and OS and lower CD44 levels in oral rinses from oral cancer cases with long-term follow-up.

## Figures and Tables

**Figure 1 nutrients-13-00372-f001:**
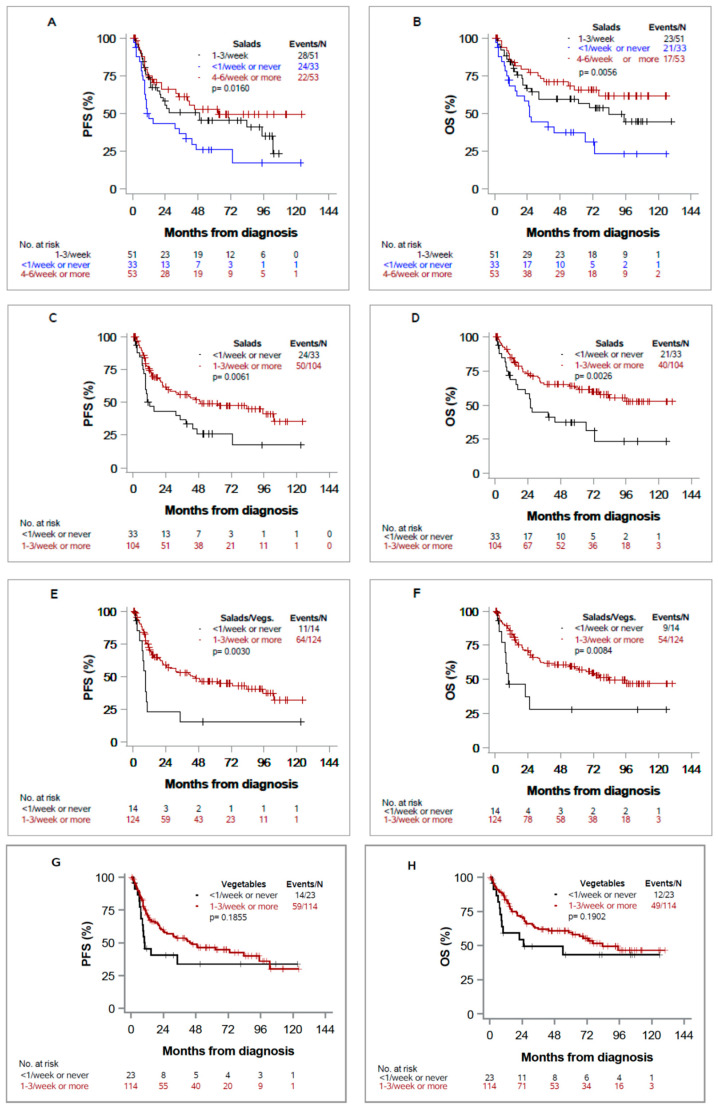
Kaplan–Meier curves for progression-free survival (PFS) and overall survival (OS) in head and neck cancer cases by intake of “salad” (plots **A**–**D**), intake of “salad or other vegetables” (plots **E**,**F**), and intake of other vegetables (plots **G**,**H**). Vertical tick marks indicate censored observations.

**Figure 2 nutrients-13-00372-f002:**
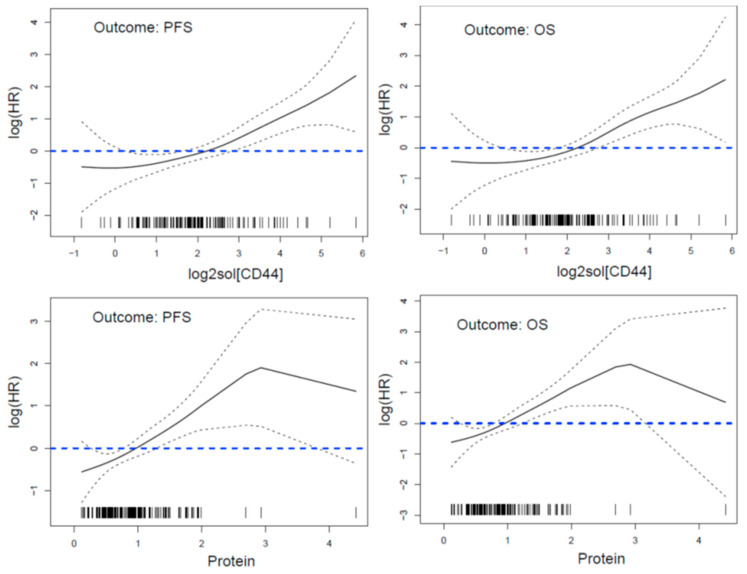
Estimated log hazard ratios (HRs) obtained from Cox regressions with restricted cubic splines functions of log2[solCD44] or protein to allow a non-linear relationship between the continuous variable and the log HR of progression or death.

**Table 1 nutrients-13-00372-t001:** Characteristics of cases and controls.

Variable/Category	Cases (*n* = 150)		Controls (*n* = 150)		*P*
	*N*	%	*N*	%	
**Age, years**					
<60	84	56.0	85	56.7	0.907
≥60	66	44.0	65	43.3	
**Gender**					
Male	121	80.7	118	78.7	0.667
Female	29	19.3	32	21.3	
**Race**					
White	123	82.6	118	79.7	0.534
Black	26	17.4	30	20.3	
Asian/Other/Missing (1 case Other, 1 control Asian, and 1 control missing)	1		2		
**Ethnicity**					
Hispanic	77	51.3	93	62.0	0.062
Non-Hispanic	73	48.7	57	38.0	
**Education**					
≤Grade 12 or GED	77	52.0	57	38.0	0.015
Some college or college graduate	71	48.0	93	62.0	
Refused/Missing	2	NA	0	NA	
**Employment**					
Out-of/unable-to work	61	40.9	45	30.0	0.048
Occupation with some income	88	59.1	105	70.0	
Refused/Missing	1	NA	0	NA	
**Income**					
$25,000 or less	86	67.2	80	58.4	0.139
>$25,000	42	32.8	57	41.6	
Refused/Missing	22	NA	13	NA	
**SES** **^1^**					
Low	100	66.7	90	60.0	0.231
High	50	33.3	60	40.0	
**Oral health score**					
Poor/Fair	80	64.0	87	58.0	0.310
Good	45	36.0	63	42.0	
Missing	25	NA	0	NA	
**Teeth removed**					
None/1 to 5	86	58.9	92	63.0	0.472
6 or more but not al/All	60	41.1	54	37.0	
**Gargle**					
Poor/Fair	38	27.5	12	8.4	<0.0001
Good	100	72.5	131	91.6	
Missing	12	NA	7	NA	
**Smoking history ^2^**					
Never	33	22.0	32	21.3	0.889
Ever	117	78.0	118	78.7	
**Drinking habits ^3^**					
Non-drinker/Mild	78	52.0	85	56.7	0.417
Moderate/Heavy	72	48.0	65	43.3	

%: Column percentage removing missing. NA: not applicable. *P*: *p*-value from chi-square test, or Fisher’s exact test. ^1^ Socioeconomic status (SES) categories high and low were defined based on income, education and employment. **High SES**: income >$25,000 or, if income was missing, “some college or college graduate” AND “occupation with some income”. **Low SES**: income ≤$25,000, or, if income was missing, low education and/or “out-of/unable-to work”. One subject missing income and education with “occupation with some income” was classified as low SES. ^2^ Smoking history: **Never**: has never smoked or has smoked <100 cigarettes a day in lifetime; **Ever**: smoked at least 100 cigarettes in lifetime. ^3^ Drinking habits: **Non-drinker/Mild**: past drinking ≤2 drinks/day or current drinking ≤2 drinks/day for 1–15 days/month; **Moderate/Heavy**: past drinking at least 3 drinks/day or current drinking ≤2 drinks/day for 16–30 days/month or at least 3 drinks/day.

**Table 2 nutrients-13-00372-t002:** Univariable and multivariable Cox regression models for progression-free survival: Effects of nutritional and other variables.

Variable	Univariable Models			Multivariable Model 1 (74 Events in 137 Patients)		Multivariable Model 2 (75 Events in 138 Patients)	
	Events/*N*	HR (95%CI)	*P*	HR (95%CI)	*P*	HR (95%CI)	*P*
Servings of 1–3/week or more vs. <1/week or never							
**Salads**	74/137	0.51 (0.31, 0.83)	0.007	0.72 (0.42, 1.23)	0.229	--	
In P16−	29/45	0.98 (0.43, 2.24)	0.971	--		--	
In P16+	8/27	0.27 (0.06, 1.16)	0.079	--		--	
In p16 = NA	37/65	0.41 (0.21, 0.82)	0.011	--		--	
**Other Vegetables**	73/137	0.68 (0.38, 1.21)	0.188	--		--	
**Salads or other vegetables**	75/138	0.39 (0.20, 0.74)	0.004	--		0.39 (0.19, 0.83)	0.014
**Carrots**	72/135	1.03 (0.65, 1.65)	0.893	--		--	
**Potatoes**	73/136	1.06 (0.54, 2.07)	0.866	--		--	
**Fruits**	78/142	0.88 (0.53, 1.47)	0.627	--		--	
**Juices**	75/137	1.54 (0.88, 2.67)	0.129	--		--	
**HPV status** P16− vs. P16+	84/149	2.39 (1.20, 4.78)	0.013	1.87 (0.78, 4.49)	0.162	2.05 (0.85, 4.95)	0.112
NA vs. P16+		1.87 (0.96, 3.64)	0.064	2.91 (1.26, 6.72)	0.013	3.25 (1.41, 7.49)	0.006
**T Stage:** T3-T4 vs. T1-T2	84/149	2.56 (1.60, 4.11)	<0.0001	2.02 (1.10, 3.70)	0.024	1.88 (1.00, 3.51)	0.049
**solCD44 ≥8.1 vs. <8.1 ***	84/149	4.57 (2.71, 7.71)	<0.0001	2.66 (1.28, 5.55)	0.009	2.41 (1.18, 4.90)	0.015
**Protein ≥1.05 vs. < 1.05 ***	84/149	2.64 (1.70, 4.11)	<0.0001	1.32 (0.73, 2.39)	0.360	1.58 (0.86, 2.89)	0.141
**Age:** ≥60 vs. < 60 years-old	84/149	2.18 (1.41, 3.37)	<0.001	1.76 (1.02, 3.03)	0.042	1.86 (1.08, 3.22)	0.026
**Race:** Black vs. non-Black	84/149	2.48 (1.49, 4.12)	<0.001	1.60 (0.79, 3.21)	0.191	1.45 (0.70, 3.03)	0.320
**Ethnicity:** Non-Hispanic vs. Hispanic	84/149	1.37 (0.89, 2.11)	0.151	1.82 (0.97, 3.44)	0.063	1.82 (0.96, 3.45)	0.066
**Gender:** Female vs. Male	84/149	0.94 (0.55, 1.6)	0.822	1.03 (0.54, 1.95)	0.940	1.14 (0.60, 2.18)	0.692
**Smoking History:** Ever vs. Never	84/149	1.64 (0.93, 2.92)	0.090	0.88 (0.44, 1.76)	0.717	1.16 (0.59, 2.29)	0.671
**Drinking Habits:** Mod/Heavy vs. Mild/Non-Drinker	84/149	1.58 (1.03, 2.43)	0.037	1.39 (0.79, 2.47)	0.256	1.34 (0.76, 2.37)	0.315
**SES:** Low vs. High	84/149	1.58 (0.99, 2.53)	0.056	1.27 (0.58, 2.80)	0.553	1.20 (0.54, 2.63)	0.656
**Oral heath score:** poor/fair/missing vs. good.	84/149	2.10 (1.23, 3.58)	0.006	1.57 (0.81, 3.06)	0.183	1.67 (0.87, 3.19)	0.122
**Teeth removed:** 6 or more/All/missing vs. 5 or less	84/149	1.82 (1.18, 2.81)	0.007	1.25 (0.68, 2.29)	0.472	1.11 (0.60, 2.04)	0.747
**Gargle:** Poor/Fair/Missing vs. Good	84/149	2.10 (1.36, 3.25)	<0.001	1.31 (0.76, 2.25)	0.338	1.30 (0.75, 2.24)	0.353

P16 = NA: not available HPV status. HR: hazard ratio. *P*: *p* value from Wald test for H_0_: HR = 1 (same risk in both groups). Bold indicates results statistically significant at *p* < 0.05. * Cutpoints were determined using SAS macro %FINDCUT, based on a minimum Wald test *p* value, searching the CD44 range 1.5 to 11 by 0.1 and protein range 0.5 to 1.7 by 0.05. Multivariable model 1 provides the adjusted effect of salad intake and multivariable model 2 provides the adjusted effect of salad/other vegetables intake on progression-free survival, with adjustment for HPV status, T-stage, solCD44, protein, age, race, ethnicity, gender, smoking history, drinking habits, SES, oral heath score, teeth removed, and gargle. Thus, -- indicates a variable not included in the model.

**Table 3 nutrients-13-00372-t003:** Univariable and multivariable Cox regression models for overall survival: Effects of nutritional and other variables.

Variable	Univariable Models			Multivariable Model 1 (61 Deaths in 137 Patients)		Multivariable Model 2 (63 Deaths in 138 Patients)	
	Deaths/*N*	HR (95%CI)	*P*	HR (95%CI)	*P*	HR (95%CI)	*P*
Servings of 1–3/week or more vs. <1/week or never							
**Salads**	61/137	0.45 (0.26, 0.77)	0.003	0.64 (0.35, 1.16)	0.141	--	
In P16−	23/45	0.67 (0.27, 1.66)	0.381	--		--	
In P16+	7/27	0.25 (0.05, 1.12)	0.069	--		--	
In p16 = NA	31/65	0.43 (0.20, 0.89)	0.024	--		--	
**Other Vegetables**	61/137	0.66 (0.35, 1.24)	0.194	--		--	
**Salads or Other** **Vegetables**	63/138	0.40 (0.20, 0.81)	0.011	--		0.40 (0.17, 0.91)	0.029
**Potatoes**	60/136	1.31 (0.6, 2.89)	0.501	--		--	
**Carrots**	61/135	0.83 (0.5, 1.38)	0.480	--		--	
**Fruits**	65/142	0.98 (0.55, 1.75)	0.956	--		--	
**Juices**	63/137	1.28 (0.71, 2.32)	0.417	--		--	
**HPV status:** P16− vs. P16+	71/149	1.85 (0.89, 3.86)	0.099	1.71 (0.64, 4.59)	0.289	1.89 (0.70, 5.16)	0.212
NA vs. P16+		1.77 (0.88, 3.58)	0.109	3.70 (1.41, 9.70)	0.008	4.48 (1.71, 11.72)	0.002
**T stage:** T3-T4 vs. T1-T2	71/149	2.55 (1.52, 4.29)	<0.001	1.85 (0.93, 3.67)	0.081	1.68 (0.84, 3.37)	0.145
**solCD44 ≥8.1 vs. <8.1 ***	71/149	5.48 (3.14, 9.55)	<0.0001	3.17 (1.43, 7.03)	0.004	3.12 (1.45, 6.69)	0.004
**Protein ≥1.05 vs. < 1.05 ***	71/149	2.79 (1.74, 4.47)	<0.0001	1.36 (0.72, 2.60)	0.347	1.61 (0.84, 3.10)	0.151
**Age:** ≥60 vs. <60 years-old	71/149	2.18 (1.36, 3.50)	0.001	2.15 (1.15, 4.03)	0.017	2.14 (1.15, 3.97)	0.016
**Race:** Black vs. non-Black	71/149	2.97 (1.74, 5.05)	<0.0001	1.62 (0.77, 3.44)	0.205	1.68 (0.78, 3.61)	0.188
**Ethnicity:** Non-Hispanic vs. Hispanic	71/149	1.88 (1.17, 3.02)	0.009	3.04 (1.51, 6.10)	0.002	2.54 (1.28, 5.01)	0.007
**Gender:** Female vs. Male	71/149	1.22 (0.70, 2.14)	0.477	1.55 (0.77, 3.13)	0.221	1.65 (0.82, 3.35)	0.163
**Drinking Habits:** Mod/Heavy vs. Mild/Non-Drinker	71/149	1.72 (1.08, 2.75)	0.023	0.85 (0.39, 1.84)	0.675	1.25 (0.58, 2.67)	0.566
**Smoking History:** Ever vs. Never	71/149	1.64 (0.86, 3.13)	0.130	2.04 (1.06, 3.92)	0.032	1.87 (0.99, 3.54)	0.053
**SES:** Low vs. High	71/149	1.54 (0.92, 2.58)	0.104	1.94 (0.81, 4.64)	0.135	1.60 (0.68, 3.78)	0.286
**Oral heath score:** poor/fair/missing vs. good.	71/149	1.89 (1.07, 3.35)	0.028	1.31 (0.63, 2.75)	0.470	1.48 (0.73, 2.99)	0.279
**Teeth removed:** 6 or more/All/missing vs. 5 or less	71/149	1.45 (0.91, 2.32)	0.117	0.97 (0.49, 1.90)	0.925	0.95 (0.47, 1.90)	0.884
**Gargle:** Poor/Fair/Missing vs. Good	71/149	2.45 (1.53, 3.92)	<0.001	1.35 (0.72, 2.53)	0.355	1.25 (0.68, 2.31)	0.476

P16 = NA: not available HPV status. HR: hazard ratio. *P*: *p* value from Wald test for H_0_: HR = 1 (same risk in both groups). Bold indicates results statistically significant at *p* < 0.05. * Cutpoints were determined using SAS macro %FINDCUT, based on minimum Wald test *p* value, searching the CD44 range 1.5 to 11 by 0.1 and protein range 0.5 to 1.7 by 0.05. Multivariable model 1 provides the adjusted effect of salad intake and multivariable model 2 provides the adjusted effect of salad/other vegetables intake on overall survival, with adjustment for HPV status, T-stage, solCD44, protein, age, race, ethnicity, gender, smoking history, drinking habits, SES, oral heath score, teeth removed, and gargle. Thus, -- indicates a variable not included in the model.

## Data Availability

Any data not presented is available upon reasonable written request.

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
