# Peer review of "Green Salad Intake Is Associated with Improved Oral Cancer Survival and Lower Soluble CD44 Levels"

_nutrients, 2021, doi:10.3390/nu13020372_

Round 1
Reviewer 1 Report
The manuscript ”Green Salad Intake is Associated with Improved Oral Cancer Survival and Lower Soluble CD44 Levels” by Elizabeth Bradford Bell et al. describes a case-control study of 150 cases and controls with the aim of examining whether the consumption of fewer servings of fruits and vegetables was associated with higher soluble CD44 (solCD44) levels. In addition, the authors examine the effects of fruit and / or vegetable intake on progression-free survival and overall survival of cases of oral cancer cases.
In general:
- The topic is interesting and the outcomes are of some clinical relevance
- The paper is technically sound
- The claims are on the whole supported by the experimental data presented
- The manuscript is clearly written and the English is sufficed
However, some issues should be addressed:
- A diet containing several portions of green salad and vegetables per week is commonly connected with a more “healthy lifestyle”. What is known about other factors like exercise and BMI of the subjects? Do patients with a healthy lifestyle have a lower risk and better outcome? In that case “green salad intake” is maybe not the best measure?
- The authors study soluble CD44 levels in the saliva. Was the elevated solCD44 level in cancer patients confirmed with CD44 expression in cancer tissue biopsies? For example, the p16 (HVP) status was examined from patient derived tissues. Where these samples CD44 positive?
- CD44 is expressed in most cell types. Cancer stem cells over-express CD44, but other cancer stem cell markers are of importance i.e., CD133 and CD24. Is the elevated expression of solCD44 associated with increase of other cancer stem cell marker?
- Measuring the expression of the standard isoform CD44s, comprising exons 1–5 and 16–20, will result in measuring all variants containing these exons, including all (splice) variants. Several CD44 splice variants have been found to be associated with cancer incidence and unfavorable outcomes. In HNSCC CD44v6 levels have been found to play an important role. Which forms of CD44 are present and associated with shorter/ longer OS and PFS?
- In Figure 1 e and f, the effect of salad or other vegetables is plotted. What is the association of other vegetables without green salad intake?
- It would be interesting to see the hazard ratios correlated to the levels of solCD44. Please provide a graph showing these correlations.
- On page 3 line 145 the authors claim that the majority of cases and controls has an age less than 60 years (56% and 56.7%, respectively). However, in table 1 (page 4) 84 patients are listed as below 60 years and 86 patients older than 60 years. Which information is correct?
Author Response
Please the response to Reviewer 1 in the attachment.

Reviewer 2 Report
Nutrition related study that investigates the relationship between tumor associated marker CD44 and head and neck-cancer along with dietary intake. Important study that adds to the body of work aiming to provide dietary advise for general longevity and specifically identify chemo-preventative and chemo-therapeutic foods and compounds. Overall this work is concise and important for the field. I have several comments pertaining to the false claims of significance and misleading wording, also new analysis is required/ needs to be better explained.
CD44 in the introduction: requires a better introduction and more detail about the isoforms of CD44.
There are no details about the immune system and immune cell expression of CD44, this is an important consideration for the study of CD44.
Line 171-177: after adjusting for covariates your significants was lost. Therefore I do not see any reason to claim that you were approaching significance. Please adjust this section and any subsequent incidences where non-significant results are discussed as they could be.
Line 190-192: once again, reaching significance indicates that NO benefit has been observed. The language used is completely incorrect. Please instead state the results clearly that there was no significant difference and thus no benefit observed.
Overall: Can you justify why you have not controlled for BMI/ weight. It is likely that people eating more salads had a lower BMI and that could be the reason for clearing HPV and better prognosis etc. Please justify this/ include the co-factor BMI in your analysis. I see only 1 short sentence mentioning this in the discussion, this is completely inadequate.
Line 206-207: Once again the wording is clearly trying to misguide the reader into thinking you have significant results when the only results observed are in cases where there would be mixed HPV + and -. Please make these results clearer so there is no confusion.
Line 290: As before, remove any claims like these. Approaching significance does not exist. There is no significant relationship in this instance.
METHODS section:
Needs to be separated into sections for each assay, the format used is not acceptable. Please see any other nutrients published manuscript for details. Moreover, details about the statistics are minimal, they need to be better explained and in much more depth including explaining what tests were used for what variables etc. Eg ‘Some analyses are based on a small number of subjects given missing data for nutritional variables’ this is vague and requires additional explanation.
Author Response
Please see the comments to Reviewer 2, starting on page 3 of the attached document.
